# The Effects of the Mediterranean Diet on Health and Gut Microbiota

**DOI:** 10.3390/nu15092150

**Published:** 2023-04-29

**Authors:** Thomas M. Barber, Stefan Kabisch, Andreas F. H. Pfeiffer, Martin O. Weickert

**Affiliations:** 1Warwickshire Institute for the Study of Diabetes, Endocrinology and Metabolism, University Hospitals Coventry and Warwickshire, Clifford Bridge Road, Coventry CV2 2DX, UK; 2Division of Biomedical Sciences, Warwick Medical School, University of Warwick, Coventry CV2 2DX, UK; 3NIHR CRF Human Metabolism Research Unit, University Hospitals Coventry and Warwickshire, Clifford Bridge Road, Coventry CV2 2DX, UK; 4Department of Endocrinology, Diabetes and Nutrition, Campus Benjamin Franklin, Charité University Medicine, Hindenburgdamm 30, 12203 Berlin, Germany; 5Deutsches Zentrum für Diabetesforschung e.V., Geschäftsstelle am Helmholtz-Zentrum München, Ingolstädter Landstraße, 85764 Neuherberg, Germany; 6Centre for Sport, Exercise and Life Sciences, Faculty of Health & Life Sciences, Coventry University, Coventry CV2 2DX, UK

**Keywords:** Mediterranean diet, gut microbiota, dietary fibre, monounsaturated fat

## Abstract

The Mediterranean Diet (MD) is plant-based and consists of multiple daily portions of vegetables, fruit, cereals, and olive oil. Although there are challenges with isolating the MD from the typical Mediterranean lifestyle and culture (including prolonged ‘social’ meals and siestas), much evidence supports the health benefits of the MD that include improved longevity, reduced metabolic risk of Diabetes Mellitus, obesity, and Metabolic Syndrome, reduced risk of malignancy and cardiovascular disease, and improved cognitive function. The MD is also associated with characteristic modifications to gut microbiota, mediated through its constituent parts (primarily dietary fibres, extra virgin olive oil, and polyunsaturated fatty acids [including ω-3]). These include enhanced growth of species that produce short-chain fatty acids (butyrate), such as *Clostridium leptum* and *Eubacterium rectale*, enhanced growth of *Bifidobacteria*, *Bacteroides*, and *Faecalibacterium prausnitzii* species, and reduced growth of Firmicutes and *Blautia* species. Such changes in gut microbiota are known to be associated favourably with inflammatory and oxidative status, propensity for malignancy and overall metabolic health. A key challenge for the future is to explore the extent to which the health benefits of the MD are mediated by such changes to gut microbiota. The MD confers both health and environmental benefits. Adoption of the MD should perhaps be encouraged and facilitated more generally and not just restricted to populations from Mediterranean regions. However, there are key challenges to this approach that include limited perennial availability of the constituent parts of the MD in some non-Mediterranean regions, intolerability of a high-fibre diet for some people, and potential cultural disconnects that juxtapose some traditional (including Western) diets with the MD.

## 1. Introduction

In recent decades, our diet has attracted much interest both scientifically and within the popular media. Whilst there are many potential explanations for such a renewed interest in diet, it is incontrovertible that obesity, and the impact of our diet on body weight, has played an important role. Obesity now affects 650 million people globally (with 1.9 billion people being overweight), with global obesity rates having tripled over the last 50 years [1]. Obesity is associated with multi-morbidity (>50 medical conditions). These include Metabolic Syndrome (Type 2 Diabetes Mellitus [T2D], Hypertension, Dyslipidaemia, and non-alcoholic fatty liver disease [NAFLD]), and chronic conditions that implicate the neurological (Benign Intracranial Hypertension), respiratory (Obstructive Sleep Apnoea [OSA]), reproductive (fertility problems and Polycystic Ovary Syndrome [PCOS]), cardiovascular (Heart Failure) and skeletal systems (biomechanical disorders) [2].

Diet forms a cornerstone in any healthy lifestyle program that is implemented either for the prevention of obesity or as part of a therapeutic strategy for weight loss (the latter often in combination with pharmacotherapies for weight loss or bariatric surgery). Amongst the milieu and profit-driven complexity of diets, there are a few notable examples that have attracted the interest of researchers, for which health-promoting effects *are* validated through rigorous scientific data published in the medical literature. One example is the Low-Carbohydrate Diet (LCD), which we have reviewed previously [3]. A further notable example is the Mediterranean Diet (MD). The MD (coined by Ancel Keys in 1960 [4]) has become one of the most studied and widely reported diets and has received an inordinate amount of attention [5]. Originating in civilisations surrounding the Mediterranean Sea, it is important to note that the MD transcends a simple diet but is also associated with lifestyles and social behaviours that typify populations living in this region [5]. Therefore, one challenge of studying the MD is to separate and untangle effects that stem from the diet itself (including, for example, the metabolic benefits of wholegrain and polyphenols consumption [6]) versus those that originate from the associated lifestyles and social behaviours.

In this concise review, we consider the definition of the MD and its main health benefits. We provide a brief overview of gut microbiota, explore the effects of the MD on gut microbiota, and consider how such changes may mediate the beneficial health-promoting effects of the MD. Finally, we explore the future of the MD, including the feasibility of its widespread adoption.

## 2. Methodology

We used PubMed as a search tool. We used the terms ‘Mediterranean Diet’ and ‘Gut Microbiota’ to provide relevant articles for our narrative review. We only used papers written in English, and although we did not restrict published papers according to their date of publication, we prioritised papers that have been published more recently. Given the concise nature of our review, the papers included here are not intended to be exhaustive, and in the case of multiple published studies, we selected the most important ones.

## 3. Definition of a Mediterranean Diet

One of the challenges of exploring the effects of the MD on metabolic health and gut microbiota is a lack of consistency within the existing literature regarding how to accurately define the MD. A relatively recent review of the literature attempted to unify a definition for the MD [7]. The authors included criteria such as definitions spanning general descriptive terms, servings of key foods, and nutrient content [7]. Based on this review, the authors defined the MD according to the following daily dietary intake:Vegetables: 3–9 servingsFruit: 0.5–2 servingsCereals: 1–13 servingsOlive oil: up to 8 servings

Regarding its energy content and macronutrient composition, the MD contains approximately 2220 Kcal per day, with 37% as fat (of which 18% is monounsaturated fat, and 9% is saturated fat) and 33 g of fibre per day. Given the controversy in the current literature regarding the definition of the MD, we align our own definition for the purposes of this review to the criteria outlined here and as proposed by Davis and colleagues [7]. The criteria outlined here reveal the MD to be a plant-based diet (replete with fibre content), including seasonally fresh, locally grown and minimally processed foods [5]. Furthermore, although the fat content of the MD is relatively high, it is important to note that most of the fat content stems from monounsaturated fat (in fact, the MD contains twice as much monounsaturated fat as saturated fat). The origin of this monounsaturated fat is the relatively large volume (up to 8 servings per day) of olive oil consumption (especially virgin and extra-virgin olive oil) [5]. This stems from the Mediterranean region being closely associated with traditional olive cultivation [5]. Finally, the MD does not include any meat-based nutrients in its defining characteristics but allows for moderate consumption of red and white meat.

The fibre-rich nature of the MD is worthy of note. We previously reviewed the health benefits of dietary fibre [8]. Based on guidelines from European countries and the US, it is recommended for adults that daily intake of dietary fibre should be between 30–35 g per day for men and between 25–32 g per day for women [9]. A review of actual dietary fibre intake for nearly 140,000 individuals living in European countries revealed this to be 18–24 g per day for men and 16–20 g per day for women [9]. Based on data from the National Health and Nutrition Examination Survey (NHANES), populations living in North America are also deficient in their fibre intake. To conclude, Western populations within Europe and the US are impoverished in their dietary fibre intake, which is about a third below the recommended level, with the implication that most of us should increase our daily fibre intake by around 50% [8]. The MD is unusual amongst most current Western diets in Europe and the US in that it contains much more dietary fibre content. As such, the MD both reflects and attests to the current recommendations for the dietary intake of fibre as an essential macronutrient for health and wellbeing.

### Health Benefits of the MD

The health benefits of the MD have been reviewed in detail elsewhere [5]. Here, we provide an overview of the main reported evidence rather than a detailed exposition of the literature. Although health benefits exist for the individual components of the MD (such as olive oil), we consider the evidence for the entire MD, given the possible additive or synergistic health-beneficial effects of these components [5]. The MD appears to be associated with a reduction in the overall risk of developing chronic disease and increased longevity. However, these associations have become blurred in recent decades due to changes in diets and lifestyles [5,10]. Here we consider the evidence for the effects of MD on health outcomes regarding Diabetes Mellitus, obesity, Metabolic Syndrome (MS), malignancy, cardiovascular disease, longevity, and cognitive function.

Diabetes Mellitus (DM): Prospective studies reveal that adherence to the MD is inversely associated with risk for the development of DM [5,11]. In a meta-analysis (including studies in both Mediterranean and non-Mediterranean populations), there was a 13% reduction in the risk of developing T2D in those who followed the MD [5]. Furthermore, in the ‘ATTICA’ cohort study, adherence to the MD was associated with a lower 10-year incidence of developing DM (reductions of 49% and 69% in men and women, respectively) [5,12]. In the ‘PREDIMED’ trial, there was a 30% reduction in the risk of developing T2D in those participants randomised to the MD versus the control group [13]. Finally, a meta-analysis of five Randomized Controlled Trials (RCTs) revealed improved glycaemic control in patients with T2D and pre-diabetes who were randomised to the MD versus control (and lower fat) diets [5,14].

Obesity: In the EPIC study on >373,000 participants followed up for a median of 5 years, those with high adherence to the MD lost a very modest 0.16 Kg in body weight but had a 10% reduced risk of developing obesity or overweight compared with participants who had low adherence to the MD [5,15]. In a systematic review of the literature (including 5 RCTs), compared with a Low-Fat Diet (LFD) and at ≥12-months follow-up, the MD resulted in greater body weight loss, although similar weight loss to other comparator diets such as the LCD [5,16]. Further evidence from a meta-analysis that included 18 RCTs revealed that compared with control diets, the MD was associated with greater reductions in visceral fat and waist circumference (primarily when diets were restricted in energy intake) [5,17]. Further data to support the favourable effects of the MD on body weight stem from the DIRECT RCT that compared the MD with restricted-calorie and control diets followed up for 2 years. Overall, mean weight loss was 4.6 Kg, and was greater for those participants assigned to the MD [5,18]. Furthermore, for the participants with DM, the MD group had more favourable changes in their fasting glucose and insulin levels [5,18]. At 4 years following the intervention, compared with the other diet groups, weight loss for the MD group was favourable at 3.1 Kg [5,18].

Metabolic Syndrome (MS): In a meta-analysis that included >33,800 prospectively observed individuals, there was a 19% reduction in the risk of developing MS for those with greater adherence to the MD (including inverse associations with waist circumference and blood pressure) [5,19]. Regarding the effect of MD on remission of MS, in a subset of the PREDIMED trial, compared with the control group, the MD groups had a greater chance of remission of MS [5,20]. In a meta-analysis that included the PREDIMED trial and others (with follow-up of 2–5 years), participants with MS who had been allocated to the MD had a 49% increased probability of remission from MS [5,14]. The metabolic benefits of the MD may be mediated, at least in part, by the prebiotic effects of oat phenolic compounds, regulation of gut microbiota and antioxidant activity [21].

Malignancy: In a large meta-analysis of cohort studies (with >3.2 M participants), greater adherence to the MD is inversely associated with malignancies (including breast, colorectal, respiratory, gastric, liver, bladder and head and neck). For those who had the greatest adherence to the MD, this was inversely associated with overall cancer mortality (RR 0.87) and all-cause mortality in survivors of cancer (RR 0.75) [5,22]. Despite these encouraging meta-analysis data based on observational studies, there are limited data from RCTs on the effects of MD on cancer development and cancer mortality [5]. In the PREDIMED study, women in the ‘MD + extra-virgin olive oil’ arm appeared to have significant protection from breast cancer development, and there was a protective trend in the ‘MD + nuts’ arm [5,23].

Cardiovascular Disease (CVD): Regarding epidemiological assessments of CVD benefits from the MD, data from a prospective cohort observation (follow-up for 20 years) on >74,800 women from the Nurses’ Health Study reveal that greater adherence to the MD associated with 29% reduction in the risk for Coronary Heart Disease (CHD), and a 13% reduction in the risk for stroke [5,24]. Regarding RCT evidence, the PREDIMED trial was designed to assess the effects of the MD on primary cardiovascular prevention in those at high risk of CVD, with a median follow-up time of 4.8 years [5]. Compared with the control group, participants assigned to the MD (supplemented with extra virgin olive oil in one group and nuts in another group) had, respectively, 30% and 28% reductions in a composite primary endpoint of myocardial infarction, stroke, and cardiovascular death [5,25,26]. Furthermore, in the secondary prevention Lyon Diet Study (with a 46-month follow-up), compared with controls, the participants randomised to the MD had a 47% reduction in their risk of having a second myocardial infarction and cardiovascular death [5,27]. A recently reported systematic review and meta-analysis of prospective cohort studies (n = 38) and RCTs (n = 3) on >9000 participants evaluated the effect of the MD on the prevention of CVD incidence and mortality. For the RCTs, compared with controls, adherence to the MD reduced the incidence of CVD by 38% (including a 35% reduction in the incidence of myocardial infarction) [5,28]. For the prospective cohort studies (comparing highest vs lowest categories of MD adherence), there were relative risk reductions for CVD mortality (21%), CHD incidence (27%) and mortality (17%), and incidence of myocardial infarction (27%) [5,28].

Regarding the MD and CVD risk, perhaps our most convincing and robust RCT evidence to date stems from the recently reported CARDIOPREV study, in which patients with established CHD were randomly assigned to either a MD or a LFD, with a 7-year follow-up period [29]. Compared with the MD, those in the LFD group had a reduced intake of vegetables, fruits, legumes, and nuts [30]. Compared with the LFD in the male sub-group, those assigned to the MD had a 33% relative risk reduction for the primary endpoint (a composite of major cardiovascular events) [29].

Longevity: In a meta-analysis of 29 prospective observational studies in >1.67 M participants, there was an overall 10% reduction in all-cause mortality for every two-point increase in adherence to the MD, with a stronger impact of the MD on mortality for participants who lived in the Mediterranean region versus those who lived in non-Mediterranean regions (HRs 0.82 and 0.92, respectively) [5,31]. Contrarily, data from some intervention studies, including the PREDIMED trial, reveal no effect of the MD on all-cause mortality [25], possibly due to insufficient power and duration of such trials [5]. The case for a positive effect of the MD on longevity is strengthened and made plausible by the observation from multiple studies that adherence to the MD is associated with a longer telomere length, suggesting that ingestion of the MD may slow the biological ageing process [5,32,33]. Furthermore, the PREDIMED [25] and CARDIOPREV [29] studies both included ‘Sofrito’, an onion-tomato sauce with high contents of ‘quercetin’ from onions and other polyphenols [34], which the participants were asked to consume frequently. Quercetin is among the ‘senolytic’ agents known to drive senescent cells into apoptosis [35]. Therefore, the senolytic effect of ingested quercetin provides another plausible mechanism whereby the MD may improve overall longevity.

Cognitive Function: Published meta-analyses suggest associations between greater adherence to the MD and better episodic memory and global cognition, and with reduced risk of developing neurodegenerative diseases and cognitive decline [5,36,37]. In one meta-analysis of 31 longitudinal studies (4–26 years follow-up), greater adherence to the MD was associated with reduced cognitive decline [37]. Furthermore, in a subset of the PREDIMED study, compared with the controls, those participants allocated to the MD had improved cognition and reduced incidence of mild cognitive impairment and dementia [5,38].

To summarise this section, the medical literature is replete with both observational and interventional human-based studies across diverse populations, with a focus primarily on metabolic outcomes but also cognitive functioning, malignancy, and longevity. Although there are some inconsistencies (probably resulting from inadequate power and duration of some of the reported RCTs), the overwhelming evidence appears to support the diverse health-promoting and life-prolonging effects of the MD. In short, there is ample evidence that adherence to the MD significantly increases the probability of living a long and healthy life and optimising wellbeing. This wealth of supportive evidence promotes the MD into a relatively small ‘premier league’ of diets that differentiates it from most diets that simply lack a robust evidence base.

## 4. Overview of Gut Microbiota

We have defined the MD and outlined its health benefits. We have previously published a detailed discussion of gut microbiota [39], and in this section, we provide an overview of this topic. This summary is a necessary prelude to the next section, in which we explore the effects of the MD on gut microbiota and how such effects may mediate the well-established health benefits of the MD. ‘Microbiota’ refers to all foreign entities (non-human) that reside on and in the human body, including the genitourinary tract, respiratory epithelia, skin and most notably, the gastrointestinal tract [39]. These include Viruses, eukaryotes such as Fungi and Amoebozoa and a multitude of prokaryotes (bacteria and Archaea) [40]. The bacteria that form an important component of gut microbiota consist primarily of four major phyla, including *Bacteroides*, *Firmicutes*, *Actinobacteria* and *Proteobacteria* [41]. Overall, gut microbiota weighs around 2 Kg, and although it outnumbers our own cells (with around 100 trillion microbes) [42], only around 1000 species of human microbiota have been identified to date [43]. The majority (around 70%) of microbes that make up gut microbiota reside within the colon [43]. Through co-evolution over hundreds of millions of years, our gut microbiota and immune system have become intricately linked and, indeed, inter-dependent. In recent years (based mainly on data from rodent studies), we have transformed our understanding of this symbiotic relationship as one of great importance for our overall health and wellbeing. Furthermore, we understand that dysbiosis within gut microbiota likely underlies susceptibility to chronic disease (including autoimmune conditions, atopies, chronic inflammatory conditions, insulin resistance and mental illness) [39]. Recent data have also revealed fascinating links between our circadian rhythm and the adequacy and consistency of our sleep-wake cycle and the circadian rhythms of gut microbiota, that in term impact on the intactness of the gut wall, hepatic and overall metabolic health [44]. These data further promote the importance of avoidance of disturbance to sleep and circadian rhythm for metabolic health.

There are complex and bi-directional pathways that link gut microbiota with the human host, most notably the ‘gut–brain axis’ (GBA), that influences the central hypothalamic regulation of appetite and metabolism. The important mediators of these pathways include the metabolic by-products of gut microbiota (including short-chain fatty acids), the release of gut-derived incretin hormones, mitochondrial function, the activity of the hypothalamo-pituitary adrenal axis, autonomic signals, immune-inflammatory pathways, and gut wall integrity [39]. Other important pathways that link gut microbiota with metabolic health include the ‘gut–liver axis’ (GLA) [45]. Within the context of both the GBA and the GLA, there are common factors that influence signalling pathways, including the intactness of the gut wall and the composition of the microbiota defined as either ‘eubiosis’ or ‘dysbiosis’ (umbrella terms that essentially define the overall status of gut microbiota as either being favourable or unfavourable to health, respectively) [39]. Within the ‘eubiosis/dysbiosis’ paradigm, important factors include diet (with varied and plant-based diets providing an optimal environment), age, physical activity, alcohol intake, sleep and circadian rhythm, and overall stress [39,44]. Furthermore, some of the weight-loss benefits of bariatric surgery may also be mediated via changes in gut microbiota [46].

### The Effects of the MD on Gut Microbiota

Diet plays a key role in the regulation of gut microbiota and is an important contributor to the establishment and maintenance of eubiosis versus dysbiosis. Accordingly, the status of our gut microbiota influences our positioning on the health/disease spectrum. In this section, we explore the key scientific evidence for the effects of the MD on gut microbiota and consider gut microbiota as a possible mediator of the health benefits of the MD. The effects of the MD on gut microbiota have been reviewed in detail elsewhere [47]. Here, we provide a summary of the current main evidence.

Diet has a major impact on gut microbiota, with estimates of almost 60% of the total structure of gut microbiota at least amenable to rapid modification in response to dietary changes [47,48]. Indeed, in most published studies, the adoption of the MD is associated with gut microbiota that is different from those associated with Western-type diets [47]. Most studies reveal that the adoption of the MD is associated with greater biodiversity of gut microbiota (increased number of bacterial species), known to be favourable for health [47]. Beyond increased biodiversity, studies show variable effects of the MD on individual species and genera within gut microbiota. Whilst the Western diet is associated with high levels of *Bacteroides* in gut microbiota, the MD is associated with the genus Prevotella and with higher levels of *Faecalibacterium prausnitzii* [47,49]. In one study in patients with Metabolic Syndrome, the MD was associated with a reduction in dysbiosis and an increase in the Bifidobacterium genera [47,50]. Furthermore, not all studies show an effect of the MD on gut microbiota, including one study in which there were no significant differences in gut microbiota following 6 months of either a MD or Western-type diet [47,51]. It seems that, at least in the short term, it is not easy to significantly influence and stabilise gut microbiota through dietary changes [47].

Beyond the effects of the MD generally on gut microbiota, it is instructive to consider the impact of the main components of the MD on gut microbiota and their associated health benefits, including dietary fibres, polyphenols, polyunsaturated fatty acids ω-3, minerals and other micronutrients, refined carbohydrates, saturated fatty acids, and trans fatty acids (summarised in Table 1).

Dietary Fibres: Dietary fibres consist of complex carbohydrates (including fermentable [mainly soluble] and non- or poorly fermentable [mostly insoluble] forms) and oligosaccharides [8,41]. Soluble dietary fibres include β-glucan from cereals (barley, wheat, oats, and rye). β-glucan improves metabolic profile (blood glucose and serum cholesterol) and facilitates anti-oxidative stress [52]. Dietary fibres (including β-glucan), such as prebiotics, influence the composition of gut microbiota [41], contribute towards the establishment and maintenance of healthy and diverse gut microbiota (eubiosis), and improve gut immunity [8,47,52]. We previously reviewed the health benefits of dietary fibre [8,53,54,55,56]. Of note, the adoption of a high-fibre diet is associated with improved insulin sensitivity (a key factor that underlies metabolic health) [57,58]. We previously showed, using data from the ProFiMet interventional trial, that the ingestion of a high cereal fibre supplement (compared with a high dietary intake of protein) is associated with a significant improvement in insulin sensitivity [8,59]. Furthermore, the ingestion of cereal fibre prevented diminished insulin sensitivity upon increased dietary protein intake [8,59]. Conversely, depletion of dietary fibre intake can have serious adverse health consequences (likely mediated via effects on gut microbiota), as exemplified by a gnotobiotic mouse model [47,60]. In such a scenario, gut microbiota uses the host muco-glycoproteins as a source of nutrients, with associated impairment in the mucosal barrier function of the colon, enabling entry of enteric pathogens and ensuing colitis [47,60].

The MD, as a fibre-replete diet, provides a minimum of 14 g of dietary fibre per 1000 Kcal [47]. Dietary fibre within the MD includes ‘microbiota-accessible carbohydrates’ (MACs), which are complex carbohydrates found in fruits, vegetables, legumes, and wholegrains. MACs can alter gut microbiota and promote the growth of species that produce short-chain fatty acids (SCFAs, such as butyrate). These SCFAs may contribute towards a reduction in malignancies (including colorectal carcinoma) and improved cardiometabolic health (including the regulation of immunity, glucose and lipid metabolism and blood pressure) known to be associated with the MD [47,61,62]. In a recent meta-analysis and umbrella review of dietary management in T2D, it was shown that when compared with lower intake, increased dietary intake of MACs improved lipid profile, glycemic control, body weight, and inflammatory markers [63]. Evidence for the direct effects of dietary MACs on gut microbiota includes a murine model in which MACs had a suppressive effect on Clostridium difficile infection [64]. However, despite the supportive evidence for MACs outlined here, it remains unclear whether changes in gut microbiota-derived SCFA are an important mediator of the health-promoting effects of a high dietary fibre intake [8,65]. Indeed, improvements in insulin sensitivity and reduction in the risk of developing T2D appear to be associated more with the intake of non-fermentable insoluble cereal fibres derived from wheat or oat extracts, as opposed to highly fermentable and soluble types of fibre that typically produce SCFAs [8,56,65]. This observation may reflect the association of intake of insoluble cereal fibres with increased faecal bulk and, therefore, microbial mass [8]. Within the fibre-abundant MD, intake of complex carbohydrates promotes the growth of *Bifidobacteria* species (*Bifidobacterium longum, Bifidobacterium breve*, and *Bacteroides thetaiotaomicron*) that are favourable for health [8,66]. The fermentable (soluble) component of complex carbohydrates is associated with increased *Bacteroides* species and butyrate-producing bacteria such as *Clostridium leptum* and *Eubacterium rectale* [8,67,68]. The non-fermentable (insoluble) component of complex carbohydrates (a fraction of the MD) reaches the colon and consists of plant cell wall polysaccharides (cellulose) and resistant starches [8,69].

Polyphenols: In addition to dietary fibre, the MD also provides an abundance of polyphenols (secondary metabolites of plants derived from tyrosine and phenylalanine), which are known to manifest anti-inflammatory and antioxidant properties [47]. In one study, it was demonstrated that polyphenols present in olive oil displayed a bactericidal effect on strains of *Helicobacter pylori* and thereby as a potential chemo-preventive agent for gastric ulcers [47,70]. In another study that compared the MD (with 40 g per day of extra virgin olive oil) with a control diet in overweight and obese participants, it was demonstrated that the MD modulates the composition of gut microbiota, including lactic acid bacteria [47,71]. It was also demonstrated that the consumption of extra virgin olive oil (an important source of polyphenols in the MD) is associated with improvements in a variety of markers of inflammatory, oxidative, endothelial, and general metabolic status. This included increases in serum adiponectin and reductions in Tumour Necrosis Factor-α (TNF-α), interleukin-6, myeloperoxidase, and 8-hydroxy-2-deoxy-guanosine [47,71]. Most dietary polyphenols reach the intestine, where they are either absorbed or undergo extensive gastrointestinal biotransformation, exerting localised interactions with the microbiota and modulating the oxidative, inflammatory and immune status of the intestinal epithelial layer [72].

Although not part of the definition of a MD, the Mediterranean culture includes the consumption of wine during meals. It is important to note that in addition to extra virgin olive oil, wine (particularly red wine) also represents an important dietary source of polyphenols [73]. In a systematic review of the human-based literature, it was demonstrated that dietary grape and red wine polyphenols modulated gut microbiota (and vice versa) in humans, contributed towards beneficial microbial ecology and provided health benefits [74].

Polyphenols have a strong ability to scavenge reactive oxygen species (through multiple hydroxyl groups) and inhibit the formation of advanced glycation end products (AGEs). Interaction of AGEs with receptors for AGEs (RAGE) enhances oxidative stress and activates inflammatory pathways that are related to neurodegenerative diseases. Therefore, dietary polyphenols can help to prevent the development of neurodegenerative diseases through the inhibition of AGE production, interaction with RAGEs, and regulation of gut microbiota and the microbiota–gut–brain axis [75].

Polyunsaturated Fatty Acids (ω-3): Although not part of the defining features of the MD outlined earlier (pertaining to a plant-based diet), the MD does contain an abundance of oily fish [47], as a key source of polyunsaturated fatty acids (PUFA) known to improve cardiovascular health [47]. In addition to fish and seafood, nuts are also a source of PUFA within the MD. Amongst the PUFAs, ω-3 (derived mainly from fish oil) has undergone many research studies and is known to reduce inflammatory processes [47,76]. Furthermore, ω-3 also helps to prevent the development of breast cancer through the induction of protumor macrophage death, mediated by reactive oxygen species [77]. In a diabetic rat model, administration of flaxseed oil (a source of PUFA in the form of α-linoleic acid) reduced Firmicutes and *Blautia* within gut microbiota [47,78]. The authors hypothesised that given the positive association of Firmicutes and *Blautia* with key inflammatory cytokines such as TNF-α, PUFA improves the inflammatory milieu (including glyco-metabolic control) through modulation of gut microbiota [47,78]. In addition to improving immuno-inflammatory status (as a forerunner of many chronic non-communicable degenerative diseases), ω-3 PUFAs also appear to improve the intestinal epithelial barrier, reducing its permeability in colitis [47,79].

The MD contains an abundance of ω-3 fatty acids that stems from the predominant fresh fruit and vegetables and oily fish [80]. However, it is also worth noting that nuts and other non-olive-based plant oils that contribute towards a typical MD also contain abundant ω-6 fatty acids [81]. Compared with ω-3, there are relatively few reported studies that focus on the effects of ω-6 on gut microbiota. However, rodent-based data reveal opposing effects of tissue ω-3 and ω-6 levels on metabolic endotoxaemia and low-grade systemic inflammation [82]. These effects can be eliminated through antibiotic treatment, suggesting their mediation through changes in gut microbiota [82]. Furthermore, elevated tissue ω-3 fatty acids promote the production and secretion of intestinal alkaline phosphatase that, in turn, induce changes in the composition of gut microbiota, reduce lipopolysaccharide production and gut permeability, and reduce inflammation and endotaxaemia [82].

Minerals and other micronutrients: The plant-based MD, with its emphasis on vegetables, fruits, and cereals, provides an abundance of minerals (such as magnesium and calcium) and micronutrients [83]. In the context of our discussion, it is important to consider the potential effects of such dietary mineral abundance of the MD on gut microbiota. Such mineral effects may occur directly or indirectly via interactions with other components of the MD. Here, we provide an overview of some key evidence from the literature to support associations between dietary mineral content (of relevance to the overall composition of the MD) and changes in gut microbiota.

Magnesium: There is some evidence that magnesium orotate may modulate the microbiome–gut–brain axis and that this, combined with probiotic supplementation, may improve some aspects of gastrointestinal and psychiatric disorders that are associated with dysbiosis [84]. In a rodent-based study, it was shown that the addition of magnesium oxide to an inulin-based diet resulted in significant changes to gut microbiota composition and caecal SCFAs [85]. In a further rodent-based study, it was shown that magnesium deficiency within the diet altered gut microbiota and was associated with altered anxiety-like behaviour [86]. Finally, a study on weaned piglets revealed that dietary supplementation with potassium magnesium sulfation improved growth performance and modulated their antioxidant capacity and intestinal immunity, at least in part related to changes in the composition of the colonic microbiota [87].

Calcium: Data from the literature reveal that a high-calcium diet favours the growth of lactobacilli within gut microbiota and is associated with the maintenance of intestinal integrity, regulation of tight-junction gene expression, and reduction in the translocation of lipopolysaccharides [88]. It is not clear whether these associations stem from dietary calcium itself or its interactions with other components of the diet [88]. Furthermore, in a rodent-based study, although the prebiotics inulin and galacto-oligosaccharides had similar effects on the modulation of gut microbiota, this was strongly dependent upon the level of dietary calcium phosphate [89]. Finally, in a study on weaned pigs, the content of calcium phosphate in the diet (in combination with oat β-glucan) was shown to influence the intestinal abundance of some fermentation end-products [90].

Selenium, Phytate, Zinc and Vitamin B6: In a human-based study, there were differences in faecal *Selenomonadales* and seleno compounds between subjects who had had an acute myocardial infarction and controls [91]. In a study on chicks, it was shown that the phytate content of peas influences iron status, brush border functionality and the composition of the gut microbiome [92]. In school-aged children, it was shown that zinc deficiency influences gut microbial diversity [93]. Finally, a rodent-based study showed that host vitamin B6 deficiency significantly alters the composition of gut microbiota and its metabolites [94].

Refined Carbohydrates (RCs), Saturated Fatty Acids (SFAs) and Trans Fatty Acids (TFAs): To complement our overview of the effects of a MD on gut microbiota from the perspective of its abundant nutrients, it is also instructive to consider this topic from an opposite vantage point. The MD is notable for its relative lack of RCs, SFAs and TFAs [83]. Whilst it is beyond the scope of this review to provide a detailed account of the associations of such components with changes in gut microbiota, here we provide a summary of the data.

RCs, SFAs and TFAs are primary components of processed foods that typify a Western diet (the antithesis of the MD) and are associated with obesity, insulin resistance and a general diminishment in gut microbial diversity [95]. In recent years, there has been an increase in the dietary intake of fructose as a RC (high-fructose corn syrup, as a substitute for sucrose in processed foods) [95]. Dietary fructose intake promotes an increase in pro-inflammatory bacteria within the gut [95]. Furthermore, dietary fructose is implicated in the regulation of intestinal permeability and, in abundance, may contribute towards the pathogenesis of the ‘leaky gut syndrome’ through dysfunction of the intestinal tight junction protein and subsequent increased levels of endotoxins, pro-inflammatory cytokines and the development of NAFLD [95,96,97]. Rodent-based studies reveal an association of high fructose diets with gut dysbiosis, including changes in the composition of Bacteroidetes and Firmicutes [95,98]. Furthermore, excessive dietary intake of sucrose can result in it being unabsorbed and reaching the colon, where it can act as a substrate for microbiota metabolism [95]. Regarding diets with an abundance of SFAs and TFAs, a proportion may escape digestion within the small intestine and reach the colon with subsequent bacterial interactions [95,99]. Dietary SFAs and TFAs induce changes in multiple gut bacterial genera and may modify bile acid metabolism through bacterial biliary hydrolases [95,100]. Furthermore, the use of SFAs and TFAs as an energy source for gut bacteria is complicated by a need for oxygen in their metabolisation in a highly hypoxic colonic environment that is dominated by strict anaerobes [95,101]. To compound this problem, diets that are high in SFAs and TFAs often lack fibre, which restricts the availability of fermentation substrates for the gut bacteria [95]. Left unmetabolised and unfermented within the colon, SFAs and TFAs can exert a bactericidal effect on cell membranes, impair gut bacterial intracellular metabolism and ultimately induce a reduction in gut bacterial load [95,102].

To summarise, based on our current understanding as outlined here, the typical Western diet that contains an abundance of RCs, SFAs and TFAs is associated with dysbiosis, restricted gut microbial diversity, leakiness of the gut wall and dysmetabolic effects. It follows, therefore, that a diet such as the MD that contains a relative lack of such unhealthy dietary components would also avoid the associated detrimental effects on gut microbiota and metabolic health.

## 5. The Future of the MD and Concluding Remarks

Across all diets, the MD is amongst the most widely studied. It is clear from an abundance of evidence that adherence to the MD improves health, wellbeing, and longevity. It is also clear that the MD is associated with the modification of gut microbiota. What is less clear is the extent to which the MD-associated changes in gut microbiota mediate the health-promoting effects of the MD. There may be other non-microbiota-related changes that mediate such effects. Our current understanding of gut microbiota stems primarily from rodent-based studies. However, we need to exercise caution when interpreting such data in the context of human health and disease, particularly given key differences in the behaviour of rodents and humans (including, for example, coprophagia in the former). An important focus for future research is to explore further the mechanisms by which the MD facilitates health-promoting effects and the extent to which human-based MD-induced modifications to gut microbiota mediate such effects. A further focus for future research is the exploration of the mechanisms by which the MD regulates brain signalling (including control of appetite and metabolism) through the microbiota–gut–brain axis, and the use of stimulus-responsive natural polysaccharide-hydrogels as biomaterials for delivery systems [103].

Regardless of the mediating mechanisms, there is overwhelming evidence to support the health-promoting effects of the MD. Therefore, an obvious question is why we do not all adopt the MD. Of course, as its name suggests, the MD is easier to follow in Mediterranean countries, in which there is an abundance of olive oil, fresh fruit and vegetables for most of the year. Therefore, one reason for the lack of widespread adoption of the MD may simply reflect a relative lack of its component parts in some parts of the world, particularly during the winter months. However, other reasons may relate to food culture. Over millennia, the MD has become deeply entrained in populations from Mediterranean countries and complements the Mediterranean lifestyle and climate. This includes eating alfresco and a reduced need for hot food. Therefore, for people from non-Mediterranean populations, there are cultural hurdles that exist. Furthermore, many people from non-Mediterranean countries may struggle with the palatability of ingesting large amounts of fruit, vegetables, and olive oil each day. For some, the large fibre content of the MD may cause gastrointestinal side effects, including abdominal bloating and flatus, that would stymie effective adherence to the MD. Indeed, multiple intervention studies reveal that compared with populations from Mediterranean countries, those from non-Mediterranean countries have reduced compliance with and/or metabolic benefits from the adoption of a MD [104,105,106,107]. Finally, of relevance during the current global cost-of-living crisis, the cost of adhering to the MD may be prohibitive for some people. Highly processed foods that are depleted of fibre and represent the antithesis of the MD remain a cheaper option compared with healthy diets such as the MD, with foods cooked from their raw ingredients [108].

Given the challenges inherent to the scientific study of the human diet, it is important to consider some potential confounding factors that may temper our enthusiasm for the MD. As alluded to above, the adoption of the MD intertwines with the culture and lifestyles of people living in Mediterranean countries. There are numerous aspects of such a traditional culture and lifestyle that may improve health outcomes, including a daily siesta, extended time for meals, the prosocial nature of mealtimes and the stress-reducing effects of such behaviours, and an abundance of sunshine. Therefore, a devil’s advocate criticism of the literature outlined in this review is that the apparent health benefits and effects on the longevity of the MD may simply reflect the associated lifestyle and behaviours of people who tend to adopt the MD rather than stem from the MD itself. Furthermore, it could be argued that individuals who adhere to the MD (particularly in non-Mediterranean countries perhaps) may be more inclined to adopt other healthy lifestyle behaviours. These arguments are valid, and it may never be possible to fully tease apart the effects of the MD per se versus other associated lifestyle and behavioural factors in the context of human-based research. However, a compelling counterargument to these concerns is that the MD has been studied across diverse populations, cultures, and geo-locations (including populations living in non-Mediterranean countries) and has demonstrated consistent health benefits, including, for example, effects on mortality, albeit with a somewhat diminished impact for those living in non-Mediterranean versus Mediterranean regions [5,31]. It is likely that at least some of the health benefits attributed to the MD may, in fact, stem from the outlined associated non-dietary health-promoting lifestyle behaviours and cultures that typify Mediterranean populations. However, it also seems incontrovertible, given the wealth of published evidence, that the MD itself has health-promoting effects regardless of the cultural, climatic, geographical, and other lifestyle behavioural contexts in which it is adopted.

In addition to its health-promoting and life-prolonging effects, widespread adoption of the MD would also have significant environmental benefits. Overall, food production accounts for up to 30% of global greenhouse gas emissions and 70% of freshwater usage [5,109]. However, there are substantial variations in the environmental footprint of individual food items, with plant foods having the lowest greenhouse gas emissions, even when processing and transportation factors are considered [5]. A further environmental benefit of the MD is the encouragement of the growth of a wide range of crops, including wild species [5], thereby encouraging biodiversity and optimised ecosystems. Indeed, the EAT-Lancet Commission, tasked with developing healthy and environmentally sustainable diets for the global population by 2050, produced targets that are very similar to a traditional MD [5,109]. Dependency on fossil fuels for the production of the MD, though, could negate its sustainability, and the use of renewable forms of energy should be a focus for the future [5]. In our current era, global warming poses a substantial threat to global ecosystems and humanity. Given the potentially huge impact on greenhouse gas emissions and freshwater usage from responsible production of the MD and its endorsement by experts in the field [5,109], it is hard not to feel compelled by the environmental arguments for the widespread adoption of the MD.

Finally, based on the overwhelming evidence and compelling arguments outlined, how can we encourage and facilitate more widespread adoption of the MD beyond populations from Mediterranean regions? (If this is nobody’s responsibility, then it is everybody’s responsibility). Here, we provide suggestions for what we need to do:Educate the populace and raise awareness of the health and environmental benefits of the MD, including in our schools and across popular and social media.Improve access to and availability of fresh fruit and vegetables, with food companies and supermarkets working together to improve consumer choice and perhaps imaginative advice on cooking tips and recipes.Governments need to ensure that healthy plant-based foods are affordable, particularly for poorer members of society in lower socio-economic groups who have been affected most by the current global cost-of-living crisis and who have a higher risk for non-communicable diseases due to poorer lifestyle (not being a choice in these people).Processed foods should contain additional plant-based fibre.Address widespread misconceptions regarding the term ‘fat’ when applied to healthy products such as olive oil to refine and optimise the public understanding of fat, including that the real culprit for weight gain and obesity is sugar and saturated fat. Despite being replete with fat, the MD is associated with significant metabolic benefits, including improved insulin sensitivity [110]. This highlights the importance of the type and composition of dietary fat regarding metabolic health [110,111], with the Monounsaturated Fatty Acids (MUFA) derived from olive oil in the MD having a favourable effect on metabolic health [47].To accept that some people may simply not tolerate a traditional MD and that, in such cases, a modified version of the MD (with a greater intake of plant-based foods or different types of oil [as in the New Nordic Diet]) may be better tolerated.To change our food culture, particularly within western societies, to move away from unhealthy convenience ultra-processed, sugar-repleted and fibre-depleted foods, towards healthy fibre-rich plant-based foods redolent of the MD.

Finally, with this societal and collective ‘to-do’ list both in mind and enacted, we should encourage and enthuse ourselves and each other to cook from raw ingredients and to re-discover the enjoyment and fulfilment of cooking and healthy eating as a prosocial, wellbeing-promoting, and nourishing activity, as our forebears have performed for aeons.

## Figures and Tables

**Table 1 nutrients-15-02150-t001:** Summary of the effects of dietary components of the MD on gut microbiota and associated health benefits.

Dietary Component of the MD	Dietary Origins within the MD	Effects on Gut Microbiota	Associated Health Benefits
Dietary fibres	Plant-based foods (vegetables, fruit, cereals)	Improved diversity; Increased *Bifidobacteria* and *Bacteroides* species, and SCFA-producing bacteria (*Clostridium leptum* and *Eubacterium rectale*)	Improved cardiometabolic health, insulin sensitivity and risk of developing colorectal carcinoma
Polyphenols	Extra virgin olive oil	Changes in lactic acid bacteria; Reduced *H. pylori*	Improved inflammatory, oxidative, endothelial and general metabolic health status; Prevention of gastric ulcer
PUFAs (including ω-3)	Oily fish Seafood Nuts	Reduced Firmicutes and *Blautia* species	Improved inflammatory and immune status; Improved intestinal epithelial barrier
Magnesium and Calcium	Fresh vegetables and fruit	Changes in caecal SCFAs; Favoured growth of lactobacilli; Regulation of intestinal tight junction gene expression	Improved gastrointestinal and psychiatric disorders; Improved growth performance
Selenium, Phytate, Zinc and Vitamin B6	Fresh vegetables and fruit	Regulation of intestinal brush border functionality and composition and diversity of gut microbiota	Possible impact on cardio-metabolic risk; Iron status
Relative lack of RCs, SFAs and TFAs	Lack of processed foods that typify a Western diet	Reduced propensity for diminishment of diversity and pro-inflammatory nature of gut microbiota	Reduced propensity for development of obesity, insulin resistance, endotoxaemia, leaky gut and metabolic dysfunction

(*H. pylori*: *Helicobacter pylori*; MD: Mediterranean Diet; ω-3: Omega 3; PUFAs: Polyunsaturated Fatty Acids; RCs: Refined Carbohydrates; SCFA: Short Chain Fatty Acid; SFAs: Saturated Fatty Acids; TFAs: Trans Fatty Acids).

## Data Availability

Not applicable.

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
