# Peer review of "The Effects of the Mediterranean Diet on Health and Gut Microbiota"

_nutrients, 2023, doi:10.3390/nu15092150_

Round 1

Reviewer 1 Report

The Mediterranean Diet (MD) has received an inordinate amount of scientific interest and published research. The MD is a plant-based diet that consists of multiple daily portions of vegetables, fruit, cereals, and olive oil. Although there are challenges with isolating the MD from the typical Mediterranean lifestyle and culture (including prolonged ‘social’ meals and siestas), much evidence supports the health benefits of the MD that include improved longevity, reduced metabolic risk of Diabetes Mellitus, Obesity, and Metabolic Syndrome, reduced risk of malignancy and cardiovascular disease, and improved cognitive function.

In addition to these well-established health benefits, it is also clear that adoption of the MD associates with characteristic modifications to the gut microbiota, mediated through its constituent parts (primarily dietary fibres, extra virgin olive oil, and polyunsaturated fatty acids [including ω-3]). These include enhanced growth of species that produce short-chain fatty acids (butyrate) such as Clostridium leptum and Eubacterium rectale, enhanced growth of Bifidobacteria, Bacteroides, and Faecalibacterium Prausnitzii species, and reduced growth of Firmicutes and Blautia species. Such changes in the gut microbiota are known to associate favourably with inflammatory and oxidative status, propensity for malignancy and overall metabolic health.

It is interesting topic and the manuscript is well organization. However, there still have some issue need to check.

1.      Please compress the abstract part.

2.      “Keywords” should be adjusted.

3.      Introduction part: the Mediterranean Diet has characteristics about wholegrain consumption (Whole grain benefit: oat β-glucan and phenolic compounds synergistically regulates hyperlipidemia via gut microbiota in high-fat-diet mice. Food & Function, 2022, 13(24), 12686-12696. Doi: 10.1039/d2fo01746f) and polyphenols(Oat phenolic compounds regulate metabolic syndrome in high fat diet-fed mice via gut microbiota. Food Bioscience. 50(2022): 101946. Doi: 10.1016/j.fbio.2022.101946) consumption increased.

4.      Metabolic Syndrome is associated with polyphenols and gut microbiota (The positive correlation of antioxidant activity and prebiotic effect about oat phenolic compounds. Food Chemistry, 402(2023): 134231.) consumption increased.

5.      The mechanism about obesity should be analyzed, such as oxidative stress and carbonyl stress. The figure about its mechanism should be provided. Thus, the MD diet should consider the metabolic activity and delivery system (Recent advances of stimuli-responsive polysaccharide hydrogels in delivery systems: A review. Journal of Agricultural and Food Chemistry. 70(21):6300-6316.).

6.      The important mediators of these pathways include the metabolic by-products of the gut microbiota should be include the gut-microbiota axis(Dietary polyphenols: regulate the advanced glycation end products (AGEs)-RAGE axis and the microbiota-gut-brain axis to prevent neurodegenerative diseases. Critical Reviews in Food Science and Nutrition. Doi: 10.1080/10408398.2022.2076064.).

7.      Dietary Fibres: the soluble fiber should be included (Recent advances of cereal beta-glucan on immunity with gut microbiota regulation functions and its intelligent gelling application. Critical Reviews in Food Science and Nutrition. doi: 10.1080/10408398.2021.1995842.)

8.      ω-3 fatty acids mainly come from fish oil, which also benefit the cancer prevention in MD diet(Consumption of the fish oil high-fat diet uncouples obesity and mammary tumor growth through induction of reactive oxygen species in pro-tumor macrophages. Cancer Research, 2020, 80(12): 2564-2574.).

The expression should be improved before it can be published. The reference should be updated.

Author Response

Reviewer 1

Comment 1.1:

It is interesting topic and the manuscript is well organization. However, there still have some issue need to check. Please compress the abstract part.

Response to comment 1.1:

Thank you for this comment. We agree with the reviewer. In response, we have truncated the abstract in the revised version of our manuscript.

Comment 1.2:

“Keywords” should be adjusted.

Response to comment 1.2:

Thank you for this comment. In the revised version of the manuscript, we have adjusted the keywords, and added ‘dietary fibre’ and ‘monounsaturated fat’ as additional terms.

Comment 1.3:

Introduction part: the Mediterranean Diet has characteristics about wholegrain consumption (Whole grain benefit: oat β-glucan and phenolic compounds synergistically regulates hyperlipidemia via gut microbiota in high-fat-diet mice. Food & Function, 2022, 13(24), 12686-12696. Doi: 10.1039/d2fo01746f) and polyphenols (Oat phenolic compounds regulate metabolic syndrome in high fat diet-fed mice via gut microbiota. Food Bioscience. 50(2022): 101946. Doi: 10.1016/j.fbio.2022.101946) consumption increased.

Response to comment 1.13:

Thank you for this comment. We agree with the reviewer. In response, we have included some additional text to the Introduction section of our revised manuscript, and the suggested additional reference.

Comment 1.4:

Metabolic Syndrome is associated with polyphenols and gut microbiota (The positive correlation of antioxidant activity and prebiotic effect about oat phenolic compounds. Food Chemistry, 402(2023): 134231.) consumption increased.

Response to comment 1.4:

Thank you for this comment. We agree with the reviewer. In response, we have included some additional text to the sub-section on ‘Metabolic Syndrome’, and the suggested additional reference.

Comment 1.5:

The mechanism about obesity should be analyzed, such as oxidative stress and carbonyl stress. The figure about its mechanism should be provided. Thus, the MD diet should consider the metabolic activity and delivery system (Recent advances of stimuli-responsive polysaccharide hydrogels in delivery systems: A review. Journal of Agricultural and Food Chemistry. 70(21):6300-6316.). The important mediators of these pathways include the metabolic by-products of the gut microbiota should be include the gut-microbiota axis (Dietary polyphenols: regulate the advanced glycation end products (AGEs)-RAGE axis and the microbiota-gut-brain axis to prevent neurodegenerative diseases. Critical Reviews in Food Science and Nutrition. Doi: 10.1080/10408398.2022.2076064.).

Response to comment 1.5:

Thank you for this comment. We agree with the reviewer. In response, we have included some additional text and references to the revised version of our manuscript. This includes additional text in the first paragraph of section 5 (‘The future of the MD and concluding remarks’), and the third paragraph of the sub-section on ‘Polyphenols’. Given that the focus of our manuscript is on the effects of the MD on health and the gut microbiota, and the advice of reviewer 2 to reduce emphasis on obesity (see comment 2.2), we have not included an additional figure on the mechanisms linking obesity with oxidative stress in the revised version of our manuscript.  

Comment 1.6:

Dietary Fibres: the soluble fiber should be included (Recent advances of cereal beta-glucan on immunity with gut microbiota regulation functions and its intelligent gelling application. Critical Reviews in Food Science and Nutrition. doi: 10.1080/10408398.2021.1995842.)

Response to comment 1.6:

Thank you for this comment. We agree with the reviewer. In response, we have added some additional text to the sub-section on ‘dietary fibres’ in the revised version of our manuscript, with the inclusion of soluble fibres. We have also included the suggested additional reference.

Comment 1.7:

ω-3 fatty acids mainly come from fish oil, which also benefit the cancer prevention in MD diet. (Consumption of the fish oil high-fat diet uncouples obesity and mammary tumor growth through induction of reactive oxygen species in pro-tumor macrophages. Cancer Research, 2020, 80(12): 2564-2574.).

Response to comment 1.7:

Thank you for this comment. We agree with the reviewer. In response, we have added some additional text to the sub-section on ‘Polyunsaturated Fatty Acids (ω-3)’ in the revised version of our manuscript. We have also included the suggested additional reference.

Comment 1.8:

The expression should be improved before it can be published. The reference should be updated.

Response to comment 1.8:

Thank you for this comment. Through addressing all the reviewers’ comments in the revised version of our manuscript, and inclusion of all the suggested additional references, the expression has been improved, and the references updated accordingly.

Reviewer 2 Report

I am satisfied that the authors reviewed and make satisfactorily explanation on the issue of MD and gut microbiota. On the overall, the paper may appeal to researchers in the filed of diet and gut microbiota.  Nevertheless, some changes should be considered to improve the manuscript. In short:

1.     Introduction: I think it is more appropriate to focus on MD and its associated health effects rather than mentioning the obesity and related diseases in the first two paragraphs. 

2.     4.1The Effects of the MD on the Gut Microbiota

-       Dietary Fibres: In this section I recommend to review the importance of MACs (microbiota accessible carbohydrates) on gut microbiota. 

-       Extra Virgin Olive Oil: In the article, MD diet components (PUFAs, dietary fiber etc. ) were examined as a subtitle. Therefore, it would be appropriate to determine both extra virgin olive oil  and wine under the polyphenols topic.  Moderate wine or its substitutes, which are the main dietary components of the MD, should also be mentioned.

-       It is recommended to explain the mechanism by which the virgin oil components improve the microbiota.  

3.     There are many similar review articles on this topic. In order to distinguish it from other articles, I suggest emphasizing the features of the MD components rather than  its medical significance as nutritional therapy, but in a much wider sense comprising lifestyle and ethnic belonging.  

Author Response

Reviewer 2

Comment 2.1:

I am satisfied that the authors reviewed and make satisfactorily explanation on the issue of MD and gut microbiota. On the overall, the paper may appeal to researchers in the field of diet and gut microbiota. 

Response to comment 2.1:

Thank you for this comment. We agree with the reviewer.

Comment 2.2:

Nevertheless, some changes should be considered to improve the manuscript. In short: Introduction: I think it is more appropriate to focus on MD and its associated health effects rather than mentioning the obesity and related diseases in the first two paragraphs.

Response to comment 2.2:

Thank you for this comment. We agree with the reviewer. In response, we have truncated the first 2 paragraphs of the Introduction section of the revised version of our manuscript, so that there is more emphasis on the MD as suggested. Given the importance of a healthy diet and lifestyle for people living with obesity, and the association of obesity with multiple dysmetabolic conditions, we believe that some mention of obesity at the start of the Introduction is important, as this sets the scene for discussion of the MD and its metabolic benefits in the rest of the manuscript.

Comment 2.3:

4.1. The Effects of the MD on the Gut Microbiota. Dietary Fibres: In this section I recommend to review the importance of MACs (microbiota accessible carbohydrates) on gut microbiota.

Response to comment 2.3:

Thank you for this comment. We agree with the reviewer. In response, we have included some additional text and references to the sub-section on ‘Dietary Fibres’ (2nd paragraph), to provide a discussion on MACs and their impact on the gut microbiota and metabolic status.

Comment 2.4:

Extra Virgin Olive Oil: In the article, MD diet components (PUFAs, dietary fiber etc.) were examined as a subtitle. Therefore, it would be appropriate to determine both extra virgin olive oil and wine under the polyphenols topic.  Moderate wine or its substitutes, which are the main dietary components of the MD, should also be mentioned.

Response to comment 2.4:

Thank you for this comment. We agree with the reviewer. In response, in the revised version of our manuscript we have changed the title of the sub-section on ‘Extra Virgin Olive Oil’ to ‘Polyphenols’. We have also included some additional text and references to this sub-section, to discuss the contribution of wine consumption in the MD.

Comment 2.5:

It is recommended to explain the mechanism by which the virgin oil components improve the microbiota.

Response to comment 2.5:

Thank you for this comment. We agree with the reviewer. In response, we have included some additional text and reference in the subsection on ‘Extra Virgin Olive Oil’ in the revised version of our manuscript.

Comment 2.6:

There are many similar review articles on this topic. In order to distinguish it from other articles, I suggest emphasizing the features of the MD components rather than its medical significance as nutritional therapy, but in a much wider sense comprising lifestyle and ethnic belonging.

Response to comment 2.6:

Thank you for this comment. We have revised our manuscript according to the suggestions of both reviewers, with additional text and references regarding the components of the MD (such as polyphenols, wine, and MACs). As such, we have addressed this comment in our revised manuscript. Given the title of our review article and the remit, we have maintained an emphasis on the medical significance and health effects of the MD. Regarding the associated lifestyle and cultural factors that associate with the MD, we already included much discussion of this in the original version of our manuscript (for example, in the third paragraph of section 5, ‘The future of the MD and Concluding Remarks’).

Round 2

Reviewer 1 Report

The author has responsed the reviewer's comment point by point. It can be acceptted in current revision.